# Environmental Factors Drive Chalcid Body Size Increases with Altitudinal Gradients for Two Hyper-Diverse Taxa

**DOI:** 10.3390/insects14010067

**Published:** 2023-01-10

**Authors:** Ning Kang, Hongying Hu, Zengqian Huang, Shungang Luo, Shuhan Guo

**Affiliations:** 1College of Life Science and Technology, Xinjiang University, Urumqi 830049, China; 2Xinjiang Key Laboratory of Biological Resources and Genetic Engineering, Urumqi 830046, China

**Keywords:** Bergmann’s rule, forewing length, Pteromalidae, Eulophidae, environmental adaptation, altitudinal clines

## Abstract

**Simple Summary:**

The size of an organism is closely correlated with its physiological and ecological characteristics and strongly influences its fitness. Bergmann’s rule, originally widely applied in homeotherms, states that individuals living in a colder environment are larger than those living in a warmer region. Similar geographical patterns are also found in various groups of poikilotherms, but are still controversial, especially in Hymenoptera. In this investigation, we found a significant upward trend in body size with increasing elevation in two tiny groups of Chalcids (Pteromalidae and Eulophidae). The temperature and precipitation play a crucial role in their size variation. This result casts light on the environmental adaptation of parasitoids.

**Abstract:**

Body size is the most essential feature that significantly correlates with insects’ longevity, fecundity, metabolic rate, and sex ratio. Numerous biogeographical rules have been proposed to illustrate the correlation between the body sizes of different taxa and corresponding geographical or environmental factors. Whether the minute and multifarious chalcids exhibit a similar geographical pattern is still little known. In this research, we analyzed morphological data from 2953 specimens worldwide, including the two most abundant and diverse taxa (Pteromalidae and Eulophidae), which are both composed of field-collected and BOLD system specimens. We examined forewing length as a surrogate of body size and analyzed the average size separately for males and females using two methods (species and assemblage-based method). To verify Bergmann’s rule, we included temperature, precipitation, wind speed and solar radiation as explanatory variables in a generalized linear model to analyze the causes of the size variation. We found that there was an increasing trend in the body size of Pteromalidae and Eulophidae with altitude. The optimal Akaike information criterion (AIC) models showed that larger sizes are significantly negatively correlated with temperature and positively correlated with precipitation, and the possible reasons for this variation are discussed and analyzed.

## 1. Introduction

Body size is one of the most remarkable characteristics of all organisms, since it is significantly correlated with the biological, physiological, and ecological traits involving longevity, fecundity, metabolic and reproduction rate, and population size [1,2,3]. Bergmann’s rule [4], a widely recognized ecogeographic rule, states that animals living in colder climates are larger than those living in a warmer regions. The geographical pattern has been broadly verified in mammals [5] and birds [6], which have broad ecological niches ranging from tropical to arctic, and their body size is positively correlated with latitude.

Bergmann’s rule was originally widely applied in homeotherms, then also verified in various poikilotherms, especially variations in arthropod sizes at both inter- and intraspecific levels along altitude or latitude gradients, but general patterns in the geographical trend of body size are still contentious. Indeed, several insect clades follow the converse Bergmann’s rule [7], such as *Onthophagus* spp.(Scarabaeidae) [8], *Dichroplus Pratensis* (Acrididae) [9], and *Paropsis atomaria* (Chrysomelidae) [10]. Conversely, the body size of *Zygogramma bicolorata* (Chrysomelidae) [11], weevils [12], moths [13], butterflies [14], stoneflies [15], and psyllids [16] generally increases with elevation. However, the body size of various insects lacks a clear relationship with altitude or latitude [17,18], and these contradictions are particularly apparent in Hymenoptera.

By summarizing the relevant literature (Table 1), we found that most of the investigations into Hymenoptera were focused on ants, bees, and other larger wasps. The geographical trends vary among species and different morphological parts, and some ant species even showed a nonlinear trend with elevation [19]. Relevant research on miniature taxa only involved Encyrtidae [20] and Mymaridae [21], and only conducted a simple comparison of body-size differences, so the knowledge of small-sized taxa is still limited. The forewing length and hind tibia length are two generally used surrogates for body length in laboratory experiments, but this method lacks validation in multiple wild species. Meanwhile, they are closely correlated with longevity, fecundity, reproductive duration, and oviposition index [22,23], which provide insights into the relationship between size and environmental factors.

The various geographical trends are generally caused by the combined effect of multiple environmental factors [24]. Based on the results of several models’ analyses, the explanatory mechanisms of these trends are usually correlated with temperature and precipitation in Diptera [25,26], Lepidoptera [13], Odonata [27,28], and Orthoptera [29]. Numerous hypotheses related to body-size control in parasitoids have been widely verified in laboratory experiments [30,31,32]. In consequence, it is critical to study these relationships under realistic natural conditions. As we all know, temperature significantly decreases with increasing elevation and is linearly correlated [33]; the most frequently verified hypotheses is the temperature-size rule [34,35], which states that a higher temperature may accelerate the development rate and result in adults with a smaller body size than those living in lower temperatures; therefore, insects living in lower temperatures tend to be relatively larger. Resources availability is also crucial to insect survival and development; insects tend to reduce their reproductive investment and increase energy storage to maintain survival under the stress of starvation [36]; thus, larger-sized insects have the potential to store more resources to resist that stress [37] and can also remain warm in cooler environments due to their lower surface-area-to-volume ratio. The wing-size polymorphism and variations are also closely related to altitude among flying insects. Wing loading and the wing aspect ratio are two common parameters for flight capability. The lower flight activity and larger wing size in a highland environment compensate for the thinner air encountered during flight [24]. However, it is still unknown whether these rules also apply to minute parasitic insects.

**Table 1 insects-14-00067-t001:** A review of the application of Bergmann’s rule in Hymenoptera insects based on the significant research. CB: Follow converse Bergmann’s rule, B: Follow Bergmann’s rule.

Taxa	Genus or Species	Bergmann’s Rule	Data Sources	Factors	Measured Treat	Reference
Apidae	*Bombus*	CB	Specimens from museum	Latitude	Thorax width	[38]
Apidae	*B. lucorum*, *B. magnus* and *B. cryptarum*	B	Specimens collected from Great Britain	Latitude	Thorax width	[39]
Apidae	*B. vancouverensis*	B	Specimens collected from the United States	Latitude	Inter-tegular span (ITS), Forewing area Community-weighted mean of ITS, relative forewing length, relative hind tibia length, hair length at mesonotum, relative hair length at mesonotum	[40]
	*B. vosnesenskii*	CB		Altitude		
Apoidea, Megachilidae, Andrenidae, Halictidae, Colletidae	*Macrotera* sp., *Lasioglossum (Dialictus)* sp., *Lasioglossum (Lasioglossum)* sp., *Halictus* sp.	B (only for Relative hair length) CB	Specimens collected from north Mexico	Altitude		[41]
Apoidea	*Andrena*, *Dasypoda*, *Halictus*, *Panurginus*	B	Data from database	Latitude	ITS	[42]
	*Bombus*, *Colletes*, *Melitta*	CB				
Apoidea	At family level	B, species larger than 27.81 mg (dry weight)	Specimens collected from Mediterranean; data from articles and museum collections	Temperature	ITS	[43]
		CB, species less than 27.81 mg				
Braconidae	At family level	CB	Literature recorded	Latitude	Body length	[44]
Ichneumonidae						
Bethylidae	*Pristocera rufa*	CB	Literature recorded	Latitude	Body length	[20]
Braconinae	At family level	CB				
Encyrtidae	*Comperiella bifasciata*	CB				
Ichneumonidae	At family level	B				
Braconidae, Ichneumonidae	At family level	CB	Specimens collected from Canada and data from BOLD system	Latitude	Hind tibia lengths	[45]
Formicidae	Ant species from 427 genera	CB	Specimens collected from the United States	Altitude/latitude	Weber’s length	[46]
Formicidae	*Leptothorax acervorum*	B	Specimens collected from Alps	Altitude	Maximum cephalic width	[47]
Formicidae	At family level	CB	Specimens collected from Australia, South Africa	Altitude	Body mass, critical thermal maximum (CT_max_)	[48]
Formicidae	*Leptothorax acervorum*	B	Specimens collected throughout Europe	Latitudes	Thorax length	[49]
Mymaridae	*Anaphes cultripennis*	B	Specimens collected from Norway	Altitude	Body length	[21]
Vespidae	*Polistes bahamensis*, *P. bellicosus*, *P. comanchus*, *P. dorsalis*, *P. exclamans*	B	Specimens from museum	Latitude	Body length	[50]
	*P. annularis*, *P. apachus*, *P. Carolina*, *P. flavus*, *P. fuscatus*, *P. metricus*, *P. dominula*, *P. aurifer*, *P. bahamensis*	CB				
Vespidae	*Agelaia pallipes*	CB	Specimens collected from Santuario de Iguaque, Colombia	Altitude	Hind femur length, mesosoma height, head width/length	[51]
		B			Forewing width/length	

To address these gaps, we primarily focused on the geographical patterns of the forewing length of Pteromalidae and Eulophidae, based on a large-scale dataset in this analysis, which consists of field-collected samples from Northwestern China and data from the BOLD system. The main objectives of this research were to analyze the relationship between morphological characteristics and elevation; verify whether the body size increases with increasing elevation, as Bergmann’s rule states; investigate the dominant environmental factors contributing to this trend, such as temperature and precipitation; and offer alternative explanations for the relationship, including the temperature-size hypothesis and the resource’s availability hypothesis. We hope to provide insights that could improve the understanding of phenotypic plasticity in the future.

## 2. Materials and Methods

### 2.1. Study Area and Specimen Measurement

This study was based on a compiled dataset of 2953 specimens, which included 1406 Pteromalids (52 genera and 81 species) and 1547 Eulophids (54 genera and 92 species), including endemically common and cosmopolitan species. More detailed information and the composition of the dataset is shown in Appendix A.

The field-collected specimens were primarily from Northwestern China (mainly covering the endemic habitats of Xinjiang, Gansu, Ningxia, and western Inner Mongolia). The area provides full-scale altitudinal gradient transect ranges from 220 m in the basin to 4336.9 m in the Altun mountain national nature reserve (Appendix A). The specimens were collected using net-sweeping and yellow pan traps (malaise traps were used at two alpine stations) during July and August from 2017 to 2020, then labeled and identified definitely to the level of genus or species following a combination of classical taxonomic guides [52,53,54,55,56], and preserved in the Insect Collection of the College of Life Science and Technology, Xinjiang University, Urumqi, Xinjiang, China (ICXU). To increase the sample quantity and geographic coverage, we selected specimens from the BOLD system (http://www.boldsystems.org/. accessed on 1 February 2021), with a complete morphological graph (with scale bar), taxonomy, and collection site, to add to the dataset. The final dataset covers 217 collection sites worldwide (sites less than 10 km apart were merged into one site). We also extracted the GPS coordinates of each point for subsequent analysis. The measured specimens finally used for analysis were distributed across all the continents except the poles (Figure 1a).

We photographed each collected specimen using a Nikon SMZ25 stereomicroscope (Nikon, Japan) in conjunction with an NIS-Elements D 4.30.00 image-collecting system using 10× magnification to capture the entire body in a single image. To avoid deviation, for accurate measurements, and to increase the reliability of the data, any severely shriveled or damaged specimens were ignored and the average value of multiple measurements for each characteristic was calculated. We measured the body length (BL) from the top of the head to the end of the abdomen, excluding the ovipositor protrusion, the mesosoma length (MEL) from the anterior edge of the pronotum to the posterior margin of the propodeum, the forewing length (FL) from the tegula proximally to the outermost point of its distal end, and the hind tibia length (HTL) (Figure 1b). The forewing was spread on the microscope slides for accurate measurement. The wing area (WA) was measured by drawing the contour of the forewing, starting and ending at the tegula (Figure 1b). After summarizing the relevant literature, we found that the wing loading (WL) and wing aspect ratio (WAR) also have a certain relationship with altitude[13,57]. Since it is difficult to obtain accurate body mass data on parasitoids, the WL was replaced by MEL^3^/WA^2^ [58], and the WAR was evaluated by FL^2^/WA [59]. All the measurements were conducted using ImageJ/Fiji 1.46.

### 2.2. Environmental Data

Based on the previous predictions and hypotheses of the size-related Bergmann’s rule, wind speed has a certain effect on parasitoids’ flight ability [60], and the lower water-loss hypothesis for larger insects in drought climates [61,62], we only considered annual mean precipitation, temperature, wind speed and solar radiation to analyze their effects on the morphological traits. The environmental variables used in our study were obtained from WorldClim-Global Climate and Weather data (version 2.1 available at https://www.worldclim.org/data/worldclim21.html. accessed on 10 April 2022.), developed by Fick and Hijmans [63]. The four environmental factors for each site were extracted using the ‘surface tool’ in ArcGIS Spatial Analyst [64], based on the geographical coordinates of the specimens used in our study; the latter two variables were obtained by calculating the average value of 12 months of data; all the digital information was acquired at the spatial level resolution of 30 arcseconds.

### 2.3. Statistical Analysis

We identified the best morphological characteristics that can be used as a proxy for body length by using an OLS regression of seven morphological parameters from males and females of Pteromalidae and Eulophidae. After establishing that the FL is more closely related to the BL than other characteristics (Pteromalidae: Cor = 0.931; Eulophidae: Cor = 0.953), we analyzed the geographical patterns of FL with altitude for both sexes, using two methods at the family level: (1). In the species-based method, the arithmetic means of FL were calculated separately for each species per location, and species with fewer than three records were excluded; a total of 2134 (1233 Eulophidae and 901 Pteromalidae) specimens were used in this analysis. (2). For the assemblage-based method, the arithmetic means of FL were calculated for all specimens from the same location, ignoring the species; some sites with fewer than three records were excluded; 2706 (1420 Eulophidae and 1286 Pteromalidae) specimens were analyzed by this method. This method aims to reflect the complete situation of populations at a specific location and is more ecologically meaningful.

The generalized linear model (GLM) is more inclusive, allowing for dependent variables to be non-normally distributed. We used the FL of the different sexes as dependent variables in the GLM for the species-based and assemblage-based methods, and the corresponding environmental factors of each site were used as the explanatory variables. A gamma distribution of errors and a log link function were used in the GLM, while the stepAIC () function was applied to avoid collinearity and simplify the models according to the Akaike information criterion (AIC) values.

We analyzed the relationship between the FL and altitude for six dominant subfamilies with the largest sample sizes. In the genus-level analysis, we accessed the relationship between the altitude (dependent variable) and the FL of 16 widely spread genera (sample size: at least 30 specimens) from Pteromalidae and Eulophidae (independent variables), taking into account the interaction between sex and altitude. We indicated the correlation structure using the corAR () function to account for the spatial autocorrelation in the data and also used the AIC value to determine the best-fit model.

The models and statistical analyzes were performed in R (version 4.0.5, R Core Team, 2021) and Excel; the data visualization was conducted by using ggplot2, ggpubr, and ggpmisc; and the model results were checked by using the ANOVA function in the “car” package [65]. The R model code refers to the code in the related articles [25,26].

## 3. Results

The regression results of the FL and HTL on the BL for females and males of 51 Pteromalidae and 48 Eulophidae species confirmed the validity of the FL as the best surrogate for body size compared to the HTL (Figure 2, Appendix A). Meanwhile, the WL was considerably negatively correlated with the FL (Pteromalidae: cor = −0.639; Eulophidae: cor = −0.604), while there was no significant correlation between the WAR and the FL (Pteromalidae: cor = 0.024; Eulophidae: cor = −0.059). The FL increased significantly with altitude and decreased with temperature in both taxa for both sexes, and in both species and assemblage-based methods (Figure 3, Table 1).

The results of the species-based method revealed a more pronounced difference between sexes for both taxa; the mean FL of females was obviously larger than that of males, indicating that larger females dominated at all altitudinal ranges. Pteromalids are generally larger than Eulophids (Appendix A, all sexes combined; mean FL of Pteromalids = 1.51 mm, mean FL of Eulophids = 1.05 mm, t. test: *p* < 0.001); the FL size frequency of both taxa demonstrated significant right skewness (Appendix A), and the frequency peaks in both sexes transferred from left to right, indicating that the proportion of individuals with larger FL tends to be greater at a higher altitude, and this was remarkably evident in the males of both taxa. The slope values showed that the trend of FL variation with altitude was more pronounced in the assemblage-based method (Figure 3).

A stepwise best subset regression, including four environmental factors and sex, was conducted. The results of the GLM model on the FL from the two methods showed that it was negatively related to sex in both taxa (males have smaller FL, *p* < 0.01), and positively correlated with the annual mean precipitation (*p* < 0.01) and annual mean solar radiation, while negatively related to the annual mean temperature (*p* < 0.001) (Table 2). The annual mean wind speed was excluded and not included in any of the models with the lowest AIC. The temperature consistently provided the greatest independent explanation for the FL variation in both methods.

Five common subfamilies (Pteromalidae: Asaphinae, Miscogasterinae, Pteromalinae; Eulophidae: Entedoninae, and Tetrastichinae) showed an obvious trend (*p* < 0.05) of the FL increasing with altitude, except for Eulophinae (*p* > 0.05) (Appendix A), mainly due to the relatively large-sized *Euplectrus*, *Cirrospilus,* and *Pnigalio* generally being distributed at low altitudes (Figure 4). However, after removing the data of these genera, the FL of Eulophinae also demonstrated an increasing trend with altitude (*p* < 0.05). The altitudinal trend of Miscogasterinae and Entedoninae is particularly significant, and the variation in *Halticoptera* (*p* < 0.001) and *Entedon* (*p* < 0.001) made the greatest respective contributions (Appendix A). Based on the analysis of the genera with a sample size greater than 20 records, we found that most of the genera had a wide altitudinal range, between 200 and 4000 m; some alpine-dwelling genera, including *Selderma*, *Callicarolynia*, *Stenomalina*, *Thinodytes*, and *Entedon*, have a larger population at higher altitudes (≥3000 m), while some genera were only distributed at lower altitudes (≤1500 m), such as *Norbanus*, *Philotrypesis*, *Dzhanokmenia*, and *Omphale*.

We used the GLS model to validate Bergmann’s rule for 16 genera with widespread distribution and large populations (distributed in the altitudinal range of 200–4000 m and ≥ 30 records). The results indicated that the FL of 11 genera significantly increased with altitude among all 16 genera, while some genera demonstrated contrasting patterns (Table 3, Appendix A). The trend of *Pteromalus* was not significant (female: *p* = 0.244; male: *p* = 0.728), especially for its males, mostly due to their comparatively larger size in Pteromalidae, and the fact that there were also several larger species found at low altitudes. There was no significant correlation between the FL of *Homoporus* (female: *p* = 0.662; male: *p* = 0.707), *Neochrysocharis* (female: *p* = 0.411; male: *p* = 0.005), and *Pediobius* (female: *p* = 0.422; male: *p* = 0.281) and altitude in general. The FL of *Chrysocharis* (female: slope = −1.71; male: slope = −1.54) showed a reverse trend with increasing elevation. All the interactions between sex and altitude were not significant.

## 4. Discussion

Much of the previous research has documented the applicability of Bergmann’s rule in multiple insect groups at the species or family level, while no general pattern has been detected in parasitoids, even for the same group, and there has been little in-depth related research on minute parasitic wasps. In this study, we examined the geographical patterns in two hyper-diverse groups of Chalcidoidea and analyzed the dominant environmental factors responsible for this variation. We found that the FL of Pteromalidae and Eulophidae increased with elevation in both sexes at the family level (a pattern that follows Bergmann’s rule), with temperature and precipitation as the main environmental drivers. 

Our results, based on the two different methods, might be explained by some of the hypotheses mentioned above. The average size of chalcids from these two taxa is larger at higher altitudes than at low altitudes. Temperature was obviously the most statistically explanatory variable in both taxa, explaining most of the variance in body size among other environmental factors in our study, and may play a more direct role. This phenomenon is also in line with numerous laboratory experiments, which show that *Pnigalio soemius* (Eulophidae: Eulophinae) raised at lower temperatures have larger FL [31], and are similar to *Eretmocerus warrae* (Hymenoptera: Aphelinidae). Larger insects usually have a smaller surface-area-to-volume ratio, which may help them maintain their thermal energy and reduce the water-loss rate [61], and also facilitate the storage of the energy needed to maintain normal activities [66]. Larger individuals are generally more fertile, which facilitates the reproduction of more offspring to maintain the population in the shorter reproductive periods in alpine regions. As stated in the starvation-resistance hypothesis, larger parasitoids are able to store more energy, such as lipids and glycogen, which are involved in egg production and other physiological mechanisms. The species’ diversity and coverage of flowering plants in the alpine area were significantly lower than those at lower altitudes, meaning that parasitoids need to restore more energy to cope with the extreme environment; thus, larger species are more suitable for survival in the alpine region. Precipitation also showed a high correlation with the FL variation; it is generally inversely proportional to elevation and is supposed to be negatively correlated with FL, as with temperature, while our results are contrary to the above speculation. Numerous investigations’ results about the effect of precipitation on body size are also controversial. Precipitation not only indirectly affects the growth of the host plants of parasitoids, and thus their population density, but also directly affects the water-loss rate of insects [67]. Therefore, precipitation may play an indirect role and interact with other environmental factors in complex ways. The geographical trend in size might be the result of interactions between various environmental factors. In addition to the factors considered above, the oxygen level (P_O2_) also declines significantly with altitude, and also has effects on the insect’s body size, combined with temperature [68]. The host identity and host plants also influence the parasitoid’s body size [69]. However, the various parasitoids analyzed in this study do not have specific host information, and the data on diet and DNA barcode also remain incomplete. This will also be a future research direction.

During the extensive field collection, we found brachypterous Pteromalidae and apterous Encyrtidae species in the alpine area (≥3500 m), while no similar species were found at low altitudes. We also found that the wing loading of Pteromalidae and Eulophidae was inversely proportional to the FL (Pteromalidae: cor = −0.639; Eulophidae: cor = −0.604) and decreased significantly (*p* < 0.001) with increasing altitude. A similar trend was also found in *Drosophila* [70], *Bumbus* [40], and moth [13], mainly because a lower wing loading is generally associated with a superior flight maneuverability, reduced energy being required for flight, and enhanced flight duration. Therefore, having this feature will help them live normally in the alpine environment. However, wing aspect ratio, also an important indicator of flight performance, did not significantly correlate with either body length or altitude in our research.

We found that the populations of *Pahcyneuron*, *Halticoptera*, and *Selderma* from the Pteromalidae, and *Entedon*, *Neochrysocharis*, and *Diglyphus* from the Eulophidae were highly dominant among other groups in the alpine region. Eleven out of sixteen genera followed Bergmann’s rule, and only the genera *Pteromalus*, *Homoporus, Neochrysocharis*, and *Pediobius* showed no significant trend, while *Charysocharis* showed a transverse trend. *Pteromalus* is one of the large-sized genera of Pteromalidae; the average FL of the *Pteromalus* specimens in our data is 1.782 mm(female), 1.35 mm(male), and the average FL of all the Pteromalidae is 1.652 mm(female), 1.422 mm(male). The larger individuals in this genus are also abundant at a lower altitude, resulting in an insignificant FL trend with elevation. Two female specimens of *Chrysocharis pubicornis* from Norway, found at elevations of 65 and 63 m, are larger than the average FL of the genus (female: 1.777 > 1.03 mm, male: 1.128 > 0.997 mm); the FL of the genus showed no significant trend with altitude when we removed these pieces of data. Such contradictory trends may also be caused by the interaction between various factors or there being less diversity in alpine species, and need further information on these genera to resolve this problem. Through the statistical analysis of numerous species, we also discovered some species with a wide distribution across altitudes, including *Asaphes vulgaris*, *Pachyneuron korlense*, *Pachyneuron aphidis*, *Diglyphus isaea*, *Diaulinopsis arenaria*, and *Neochrysocharis formosus,* etc., which will provide objects for further study of the adaptation mechanism.

In our research, the geographical trends in the size of two abundant chalcids were analyzed along a wide range of altitudinal gradients. The most seminal result is that the alpine region tends to harbor larger chalcids (following Bergmann’s rule) than the lower altitudes. Temperature and precipitation are the dominant environmental drivers of this variation, and the results indicate that temperature may be more directly related to body-size variation with altitudinal gradients than precipitation. Other hypotheses, such as resource availability, can also be used to explain the trend. The combination and interaction of these factors play an important role in the formation of body-size patterns. Additional information, such as genetic data (a test of whether body-size response to altitude shows a phylogenetic signal or there are positive selected genes) and more data on different species is needed to help further understand the mechanisms that cause body-size variations. Our results may provide a cornerstone for relevant future work.

## Figures and Tables

**Figure 1 insects-14-00067-f001:**
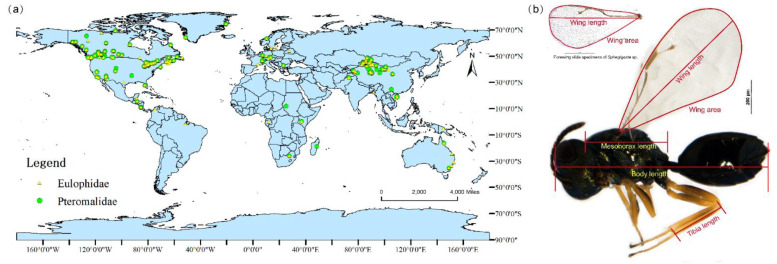
(**a**) Geographic distribution of Pteromalidae and Eulophidae specimens used in this analysis. (**b**) Morphometric measurement illustrations. Upper: microslide specimen of male forewing of *Sphegigaster stepicola* and the scheme of wing length and wing area measurement. Lower: body of female *Pachyneuron korlense* and the scheme of body length, mesothorax length, tibia length, wing length, and wing area measurement.

**Figure 2 insects-14-00067-f002:**
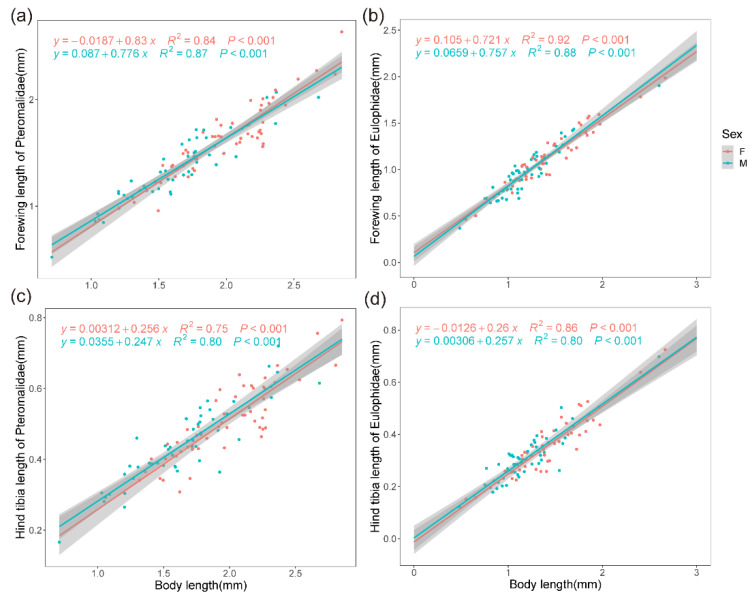
Forewing length (FL) and hind tibia length (HTL) of Pteromalidae (n = 51 males and females of the same species) and Eulophidae (n = 48 males and females of the same species) in relation to body length (BL). (**a**) Correlation between FL and BL of Pteromalidae. (**b**) Correlation between FL and BL of Eulophidae. (**c**) Correlation between HTL and BL of Pteromalidae. (**d**) Correlation between HTL and BL of Eulophidae. All *p* < 0.001. Green circles: male; red circles: female. For detailed comparison of morphological characteristics, see Supplementary Material, Appendix A.

**Figure 3 insects-14-00067-f003:**
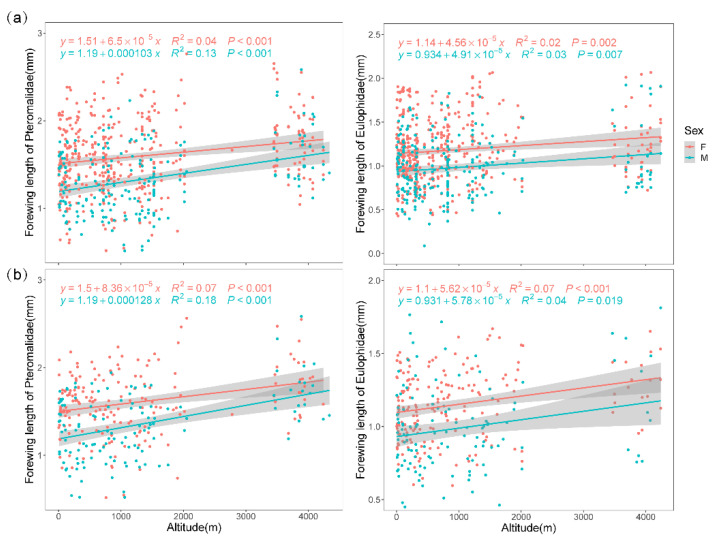
Forewing length of Pteromalidae and Eulophidae in relation to altitude based on two methods: (**a**) species-based method, (**b**) assemblage-based method. Green circles: male; red circles: female. See Supplementary Material, Appendix A for detailed frequency histograms of four different elevation ranges.

**Figure 4 insects-14-00067-f004:**
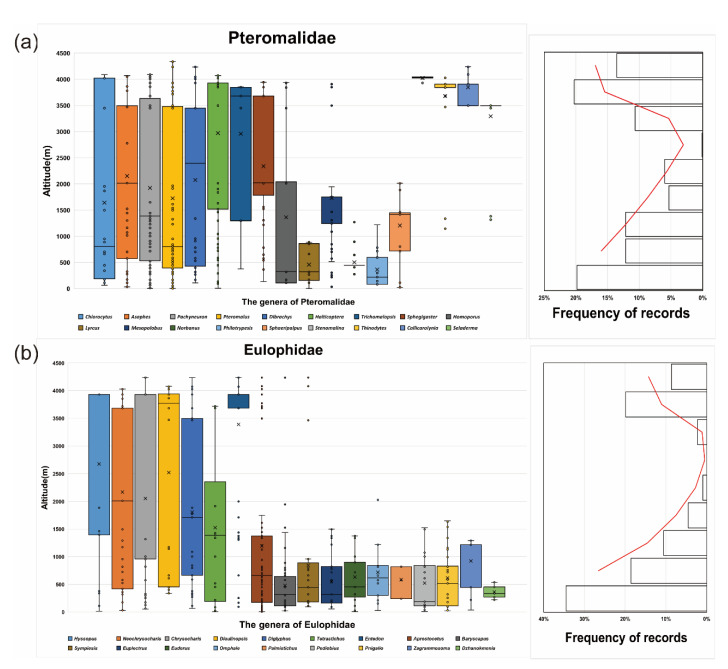
Altitudinal distribution of Pteromalidae (**a**) and Eulophidae (**b**), and frequency of occurrence records by altitude. Box and whisker plots show the median (cross symbol), the 25th and 75th percentile (bottom and top of the box), the minimum and maximum values (lower and upper whiskers), and outliers (circles). The red line shows the frequency distribution of altitudinal values in the assemblage-based data.

**Table 2 insects-14-00067-t002:** The generalized linear model (GLM) regression of relationships between FL of Pteromalidae and Eulophidae and environmental variables. AMTEM: annual mean temperature (°C), AMPRE: annual mean precipitation (mm), AMSR: annual mean solar radiation (kJ/m^2^), AMWS: annual mean wind speed (m/s). (** </ =0.01; *** </ =0.001).

Species-Based Method	Regression Coefficients	SE	95% Confidence Interval	*p*-Value
Pteromalidae				
SexM	−0.1431	0.0334	−0.208, 0.0778	2.83 × 10^−5^ ***
AMTEM	−0.0205	3.417 × 10^−3^	−0.0291, −0.0158	3.59 × 10^−8^ ***
APRE	1.449 × 10^−4^	6.262 × 10^−5^	1.253 × 10^−4^, 3.675 × 10^−4^	0.000116 ***
AMSR	3.051 × 10^−5^	8.025 × 10^−6^	−1.046 × 10^−6^, 3.003 × 10^−5^	0.0722
Eulophidae				
SexM	−0.118	0.0321	−0.018, −0.05	0.000262 **
AMTEM	−0.01756	2.977 × 10^−3^	0.023, −0.0178	7.3 × 10^−9^ ***
APRE	1.305 × 10^−4^	3.121 × 10^−5^	6.894 × 10^−5^, 1.921 × 10^−4^	3.5 × 10^−5^ ***
AMSR	1.138 × 10^−5^	6.059 × 10^−6^	−4.41 × 10^−7^, 2.32 × 10^−5^	0.061
**Assemblage-based method**				
Pteromalidae				
SexM	−0.1886	0.0273	−0.242, −0.135	2.84 × 10^−11^ ***
AMTEM	−0.0205	2.636 × 10^−3^	−0.0256, −0.0154	1.33 × 10^−13^ ***
APRE	1.499 × 10^−4^	2.773 × 10^−5^	8.945 × 10^−5^, 0.0002	3.27 × 10^−7^ ***
AMSR	3.051 × 10^−5^	6.503 × 10^−6^	1.807 × 10^−5^, 4.298 × 10^−5^	4.13 × 10^−6^ ***
Eulophidae				
SexM	−0.1526	0.02768	−0.2068, −0.0983	7.47 × 10^−8^ ***
AMTEM	−0.0149	2.595 × 10^−3^	−0.0200, −9.792 × 10^−3^	7.36 × 10^−6^ ***
APRE	1.267 × 10^−4^	2.777 × 10^−5^	7.079 × 10^−5^, 1.8309 × 10^−4^	2.38 × 10^−8^ ***
AMSR	1.136 × 10^−5^	6.031 × 10^−6^	−4.274 × 10^−7^, 2.317 × 10^−5^	0.0607

**Table 3 insects-14-00067-t003:** Coefficient and ANOVA of the best fit GLS models on FL of 16 genera from Pteromalidae and Eulophidae (CE = coefficients, n = number of specimens). * denotes level of significance (* </ =0.05; ** </ =0.01; *** </ =0.001).

	CE	χ^2^	*p*-Value		CE	χ^2^	*p*-Value
**Pteromalidae**				**Eulophidae**			
*Halticoptera* (n = 221)				*Diglyphus* (n = 98)			
Altitude	2.41 × 10^−4^	17.9192	<0.0001 ***	Altitude	8.5 × 10^−5^	20.6508	<0.0001 ***
SexM	0.01	1.5968	0.2077	SexM	−0.026	1.1148	0.2937
Altitude × Sex	−1.5 × 10^−6^	2.3774	0.1245	Altitude × Sex	−1.27 × 10^−5^	0.1044	0.7474
*Pachyneuron* (n = 162)				*Diaulinopsis* (n = 80)			
Altitude	4.28 × 10^−5^	19.1234	<0.0001 ***	Altitude	6.85 × 10^−5^	53.949	<0.0001 ***
SexM	−3.96	16.3063	0.0001 ***	SexM	−0.01	30.326	<0.0001 ***
Altitude × Sex	1.08 × 10^−3^	2.0679	0.1524	Altitude × Sex	−4 × 10^−5^	10.134	0.0021 **
*Sphegigaster* (n = 71)				*Hyssopus* (n = 30)			
Altitude	1.81 × 10^−5^	16.2064	0.0001 ***	Altitude	8.16 × 10^−5^	39.0975	<0.0001 ***
SexM	−0.58	16.1733	0.0001 ***	SexM	0.227	1.0853	0.3071
Altitude × Sex	1.45 × 10^−4^	3.7199	0.058	Altitude × Sex	5.43 × 10^−5^	1.5337	0.2266
*Asaphes* (n = 64)				*Chrysocharis* (n = 107)			
Altitude	2.13 × 10^−5^	7	0.0104 *	Altitude	−1.7 × 10^−6^	1.354	0.2473
SexM	−0.358	27.5398	<0.0001 ***	SexM	−0.061	2.433	0.1219
Altitude × Sex	8.48 × 10^−5^	8.8015	0.0043 **	Altitude × Sex	1.6 × 10^−6^	0.002	0.9641
*Pteromalus* (n = 145)				*Entedon* (n = 150)			
Altitude	4.84 × 10^−5^	0.7818	0.3777	Altitude	2.02 × 10^−4^	108.014	<0.0001 ***
SexM	−3.885	36.9243	<0.0001 ***	SexM	−0.015	2.682	0.1037
Altitude × Sex	−4.9 × 10^−5^	1.4112	0.2364	Altitude × Sex	−1.39 × 10^−5^	0.089	0.7656
*Homoporus* (n = 30)				*Neochrysocharis* (n = 115)			
Altitude	7.69 × 10^−5^	3.623	0.0681	Altitude	2.7 × 10^−6^	1.0725	0.3026
SexM	−0.6578	73.692	<0.0001 ***	SexM	−1.62	7.7498	0.0063 **
Altitude × Sex	−1.24 × 10^−4^	1.878	0.1823	Altitude × Sex	3.62 × 10^−5^	3.4063	0.0676
*Selderma* (n = 33)				*Pediobius* (n = 30)			
Altitude	3.5 × 10^−4^	31.4179	<0.0001 ***	Altitude	1.18 × 10^−4^	0.2745	0.6047
SexM	0.1187	7.2794	0.0115 *	SexM	−0.397	7.1793	0.0126 *
Altitude × Sex	−1.15 × 10^−4^	1.497	0.231	Altitude × Sex	8.37 × 10^−5^	0.1210	0.7308
*Sphaeripalpus* (n = 34)				*Aprostocetus* (n = 356)			
Altitude	3.5 × 10^−4^	12.5613	0.0012 **	Altitude	6.23 × 10^−5^	20.992	<0.0001 ***
SexM	0.0048	1.8255	0.1859	SexM	−0.117	19.163	<0.0001 ***
Altitude × Sex	−2.82 × 10^−5^	0.067	0.7973	Altitude × Sex	−2.13 × 10^−5^	0.739	0.3904

## Data Availability

The data presented in this study are available on request from the corresponding author, and also available in supplementary material here.

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
