# Peer review of "Environmental Factors Drive Chalcid Body Size Increases with Altitudinal Gradients for Two Hyper-Diverse Taxa"

_insects, 2023, doi:10.3390/insects14010067_

Round 1

Reviewer 1 Report

The manuscript “ Environmental Factors Drive Chalcid Body Size Increases with Altitudinal Gradients for Two Hyper-diverse taxons “ explores how morphological trends of Chalcids is driven by environmental drivers, mostly temperatures and precipitations. This paper is tackling an interesting question, because body size trends at large scales are indeed poorly studied in many tiny insect species. The biological questions they want to answer is thus clear. However, I have major concerns on several points. First, the manuscript is poorly written. I stopped fixing all the problems about wording and orthograph by the introduction, because every two sentences, there were typos, grammatical issues, etc. The paper would need to be fully revised, by a proofreading service probably, because there is a lot to be fixed. There are also quite a lot of problems in the tables and figures, I highlighted them below. Many methodological points are not clear to me: the calculation of some variables, some analyses that are present in the results but not in the Material and Methods, etc. Throughout the manuscript, there are some statements that are not incorrect, or imprecise, I thus also have some concerns about the deep understanding of the authors, about body size trends at this geographical scale. I detailed these points below. 

Abstract

Line 10. Influence -> influences

Line 15. Rule -> role

Line 16. “provides a relatively new basic thoughts for investigating” -> “shades light on “

Line 18. “correlated” -> “correlates”

Line 22. “rarely” -> “poorly”

Line 26. “to verify the Bergmann’s rule, temperature,” -> “To verify the Bergmann’s rule. We included temperature, “

Line 28. Remove the word “conspicuous”.

Line 31-32. Delete “Which are the dominant driving forces of these geographical patterns”.

Line 31-32. I would have like a sentence to summarize the discussion/conclusion, at the end of the abstract.

Introduction

Line 39. Reproduction rate and fecundity are basically life history traits, so it does not make sense to cite “life history” and these two traits. Either remove “life history” or the two other traits from this sentence.

Line 40. I am not sure you can say it is the oldest ecogeographical rule because Golger’s rule, another geographical rule, has been stated in 1833, so before Bergmann’s rule.

Line 40-44. This sentence is way too long and needs to be split.

Line 47. “and general patterns” -> “but general patterns”

Line 48-49. “ Various geographical patterns of insects have been found to converse with Bergmann’s Rule “ -> “Indeed, several insect clades follow the converse to Bergmann’s rule “

Line 49. Something does not make sense here. You first state that Chrysomelidae are following the converse to Bergmann’s rule as if it was something general. Then you say that some Chrysomelidae are following the Bergmann’s rule. Add more nuance in the first sentence.

Line 50-51. Delete “ more species of “

Lines 55-58. Overall, in your manuscript, your sentences are way too long. I suggest to go through your manuscript again, and split this kind of long sentence in two separate ones.

Line 59-60. I don’t understand what you mean by “ a simple comparison of individual species’s size”.

Line 61. “due to” -> “because”

Lines 61-63. I am not sure to understand this part: “ due to they are closely correlated with longevity, fecundity, reproductive duration, and oviposition index “. Do you mean that, because hind tibia and wing length are linked to all these traits, they can be used as a surrogate of body size? This argument seems poor to me. I guess that, if they are used as surrogate of body size, it’s because studies have shown that they were actually linked to body mass, or to the total length of the individual maybe.

Lines 68-70. What hypotheses are you talking about? Hypotheses in the wild that are verified in controlled experiments (which ones?)?

Lines 70-75. This sentence is really long and the English is poor. Overall in this manuscript, the writing and the English are quite poor and will need an important and careful revision. I suggest using a proofreading service to improve the overall quality. From this point, I’ll stop rewriting and fixing the problems in sentence structure/wording (because they exist in most of the sentences). I cannot spend time on re-writing the paper. I will now focus on the scientific points.

Line 79-80 “ due to the relatively lower surface area” -> “due to a lower surface-area-to-volume ratio“.

Line 87. Here and elsewhere in the manuscript. Check that Bergmann (with two “n”) is spelled correctly.

Table 1. There are some approximation with this Table 1. Some references you cite don’t only compute one analysis, but much more. By going through some of these papers, you’ll notice, for example, that reference 37 is not only finding a “Bergmann’s trend”, but a lot of taxa don’t follow any trend, or even follow the converse trend. I suggest being much more careful with the claims you’re doing here, and go in the details of each trend in the papers you are citing, before doing general statement.

Line 92-93. I don’t understand what you have verified by stating this: “ we mainly verified the following speculations: the morphological characters in relation to body length and elevation, (…)”.

Lines 94-95. Precipitation has nothing to do with the temperature-size rule, it is not a potential mechanism of this rule.

Material and Methods

Lines 113-116. If I get it well, you collected specimens in the BOLD system, from any year (while it’s only during the time span 2017-2020 for the field specimens). Do you think it could affect your results? For example, several papers showed that during the last decades, body size of some insects have shifted drastically. Isn’t there any risk that, inside a family, the body size is not only shifting across space but also along time period?

Lines 132-134. The expected changes in body size are probably not that huge, so every measurement bias you can avoid should be done carefully. For example, in my opinion, flattening a wing with microscope slides should have been done for every wing, even when you think it is flat by simply looking at it. Indeed, flatten the wing can always affect the measurement and give you more precise numbers. Do you think it could affect your results?

Lines 124-138. I think it is the first time you mention all these measurements (up to here, you mentioned wing size and body size, and briefly tibia length). Why did you choose to measure all these morphological traits? Could you justify these choices in this paragraph?

Lines 147 – 151. Have you extracted the environmental data from the corresponding year for every specimen?

Lines 156 – 160. Here, I would like to know how many individuals per species and per location you had, to calculate these averages. Did you have enough individuals per species and per location to be able to calculate a mean per species and per location? Moreover, I am not sure what is the calculation you used to have the mean FL of a species. Do you calculate a mean value for a species per location, then, average the mean value among all locations for a same species? What was the calculation for the mean FL value for a genus?

Line 160-161. What is considered a sampling site? Is it a municipality? Are you merging data from different years? Again, how many specimens did you have for each site? Have you excluded sites where you had too few specimens?

Lines 170-175. Why did you only assess the impact of altitude (and sex) here, and not the impact of the other environmental variables?

Line 171. At least 30 what? Specimens? Species?

Results

Figure 2. please increase the size of the font for the numbers on the axes, as well as the size of the equation of the model, r-squared and p-value, they are too small. Same for the size of the legend (Sex, M and F). Same for figure 3. The size of the font should at least be 9.

Legend of Fig. 2. You state there is a negative correlation for fig. 2d. I see a positive correlation.

Line 195. Significantly smaller than females? It is not clear.

Figure 3. The data points should be included on the graph so that we can better understand how well the data fit with the model, or if the trend seems to be driven by a subset of points.

Line 225. Have you ever mentioned that you’ll assess the trend at the subfamily level in the M&M? I don’t think I have seen any mention of that analysis.

Lines 226 – 237. There is no statistics provided for this paragraph in the main text, they are only mentioned in the supplementary figures, I would like to have some statistics in the main text so that the reader does not have to check them in the supplementary figures.

Figure 4. Boxplots by themselves are not super informative. Could you please include the raw data (i.e. data points) in these box plots, so that we can better visualize how the data are dispersed? Also, it would be great to visualize on these figures where are the significant differences, as you have a lot of boxplots, you could add letters at the top of the boxplots to show where are the significant differences.

Table 3. The second column cannot be the degree of freedom. For example, for Halticoptera, how could they be something like 0.000241 ?  Also, what is the number under brackets after the genus name? The number of data? It should be clearly stated.

Discussion

Line 274. I would avoid spending too much time on the temperature-size rule and would focus on the Bergmann’s rule. Indeed, the T-S rule is something quite precise: it is a plastic response (you don’t know if what you observed is plastic or not in your study) observed at different developmental temperatures (you did not measure precisely the developmental temperature), this is why this rule is mostly used in controlled conditions. At a huge geographical scale like in your study, with plenty of confounding factors, I would mostly focus on the ecogeographical rule: the Bergmann’s rule. You can briefly state that in controlled conditions, there is some similar results about the impact of temperature, but not be too confident that this is the temperature-size rule that you observe in your manuscript.

Line 278. I am wondering if, for minuscule insects like the one you observed, does the surface-area-to-volume ratio really has any impact of heat conservation/water loss? Do you have any references showing that it has an impact in such small invertebrates?

Line 286-287. Is there actually any evidence that precipitations are leading to body size trends in insects? You have observed that precipitation can be significantly correlated to body size in your data, but could it just be because it is also correlated to other variables like temperature, or do you think precipitation has an actual effect on body size (what would be the potential mechanism if you think so?)?

In the discussion: there are few genera of both Pteromalidae and Eulophidae that are only found in high elevation (if I refer to your figure 4). Are these genera particularly large? Could they be the one driving the trends at the family level? If you remove these few genera from the family level analysis, do you still have a significant relationship?

Line 323. How would genetic data help? Could you at least give an example under brackets, it is unclear to me?

Author Response

Thank you so much for every detail you have raised to improve this manuscript, and for giving us the opportunity to revise. I very appreciate the effort and time you put into this article, and I feel very sorry for such numerous errors that were ignored before. All the problems you pointed out about the writing have been revised or rewritten in the article directly, including adding a summary sentence for the discussion at the end of the abstract, splitting the long sentences, correcting the wrong and unclear statements et al.(in the abstract and introduction). Beyond that, the manuscript has been sent to the editing service in the MDPI for extensive english revision, and every detail was checked and added to the modification. We have studied all your comments carefully and have made a conscientious correction. All the writing problems in lines 10-87 have been revised directly in the text according to your suggestions, and every revised portion is marked in the paper. We would like to show the content of responses to problems on the scientific points in the attachment.

Reviewer 2 Report

It is very interesting and useful research in overall. I have only few comments on this manuscript.

1. Can authors provide the reference for the statement line 68-70?

2. When measuring body length how do you ensure that all specimens are not affected by shrinkage or other factors that could affect length measurement and if the body length can be measured why other morphological characters as a proxy for the body length are needed?

3. Can the authors explain a bit more about how the edge of wing can be used to calculate wing area? For example, wing area can be measured using the centroid size from morphometric approach or other methods.

4. Line155: Authors should provide the reason why FL is used as a proxy for body length.

5. For the species-based method, do the authors pull the data of each species in the same genera and then analyse them together or analyse each species separately? I would suggest the author to clarify this point.

Author Response

Thank you so much for the time and effort that you have put into reviewing the original version of the manuscript. Your suggestions have enabled us to improve our work, we have revised the manuscript base on all your comments and english polishing was done in MDPI, all the revised portion are marked in red in the paper. Please see the attachment for detail infromation. 

Round 2

Reviewer 1 Report

I recognize that the manuscript “ Environmental Factors Drive Chalcid Body Size Increases with Altitudinal Gradients for Two Hyper-Diverse Taxa “ has been deeply improved. Most of my concerns have been taken into account. However, before the publication of this manuscript, I would like to highlight again that there are problems in the writing of this manuscript, this task has not been done with enough care. Below, you will find three examples of sentences where there are still typos, or incorrect constructions of the sentences :

Lines 11-13. “ Bergmann’s rule, originally widely applied in homeotherms originally, states that individuals living in colder environment are larger than those living in a warmer region.”

-- the word “originally” appears two times, one of them should be removed

Lines 324 – 327. “ Generally, larger insects usually with smaller surface-area-to-volume ratio, which may help them maintain their thermal energy and reduce water loss rate[61], and also facilitates the storage of more the energy needed to maintain normal activities[66]. “

-- Something is lacking in this sentence, a verb maybe, like “Generally, larger insects usually have smaller surface-area-to-volume ratio, xxx”. Also, you don’t need to use “generally” and “usually” in the same sentence.

Lines 386 – 389. “ Temperature and precipitation are dominant environmental drivers of this variation, and the results indicated that temperature may be more directly related to body size variation, altitudinal gradients. “

-- “altitudinal gradients” is not linked to the sentence…

You can still find this kind of mistake throughout the manuscript, and this is a problem for the publication. Either the proofreading has not been done correctly, or the proofreading service has not read the last version of this manuscript.

Author Response

Thank you so much for the time and effort that you have put into improving the manuscript. Every suggestion you have made is very valuable to us. We have revised the manuscript base on all your comments, and all the revised portion is marked in red in the paper. In addition, we reviewed the article verbatim, and corrected many previously unnoticed errors directly in the text, such as adjusting part of the word order, delete words with repeated meaning, revised some misquoted literatures, and modified the format of references. 
